# Endometriosis, Oocyte, and Embryo Quality

**DOI:** 10.3390/jcm12134186

**Published:** 2023-06-21

**Authors:** Sania Latif, Ertan Saridogan

**Affiliations:** 1Reproductive Medicine Unit, University College London Hospital, London NW1 2BU, UK; sanialatif@nhs.net; 2Institute for Women’s Health, University College London, London WC1E 6HU, UK

**Keywords:** endometriosis, in vitro fertilization, oocytes, embryonic structures

## Abstract

Endometriosis is a common finding among women with infertility, and women who are diagnosed with endometriosis are almost twice as likely to experience infertility. Mechanisms by which endometriosis causes infertility remain poorly understood. In this review, we evaluate the current literature on the impact of endometriosis on oocyte and embryo quality. The presence of endometriosis evidently reduces ovarian reserve, oocyte quality, and embryo quality; however, this does not appear to translate to a clear clinical impact. Analysis of data from large assisted reproduction technology registries has shown that women with endometriosis have a lower oocyte yield but no reduction in reproductive outcomes. There is a need for future studies in the form of well-designed randomized controlled trials to further evaluate the role of surgical and medical treatment options in women with endometriosis undergoing assisted conception.

## 1. Introduction

Endometriosis is a common finding among women with infertility, and women who are diagnosed with endometriosis are almost twice as likely to experience infertility [1]. The three main subtypes of endometriosis include peritoneal, deep infiltrating, and ovarian, which frequently co-exist [2]. Mechanisms by which endometriosis causes infertility remain poorly understood. Proposed theories include mechanical distortion of pelvic anatomy, impairment of gamete and embryo transport, impaired folliculogenesis, reduced oocyte quality and embryo quality, altered immune and endocrine function, and dysregulation of hormonal and cell-mediated functions involved in endometrial receptivity [3,4,5,6,7,8,9,10].

In vitro fertilization (IVF) is an established management option for endometriosis-associated infertility [11,12]. Several studies describe a negative effect of endometriosis on the reproductive outcomes of women undergoing IVF [13,14,15,16] in contrast to others which show no difference in outcomes [17,18,19,20,21,22]. The reasons for these observed differences and whether endometriosis affects oocyte quality and embryo quality have not yet been clearly determined. In this review, we evaluate the current literature on the impact of endometriosis on oocyte and embryo quality and the management strategies which have been considered.

### Oocyte and Embryo Quality Assessment

In 2011, an expert panel convened to develop a set of recommendations to define the minimum criteria for grading oocytes and embryos [23]. Following scientific and technological advances in the field, these consensus points are currently undergoing review to provide updated recommendations. The development of the human embryo is directly influenced by the nuclear and cytoplasmic maturation of the oocyte [23]. Oocyte morphology is a proposed marker of oocyte quality [24], and its assessment is based on intracytoplasmic features of the oocyte such as the homogeneity of cytoplasm, presence of vacuoles, aggregation of smooth endoplasmic reticulum and extracytoplasmic features such as first polar body morphology, perivitelline space size, zona pellucida defects, and shape anomalies [23]. Ideal oocyte morphology consists of a circular structure with a uniform zona pellucida, uniform translucent cytoplasm free of inclusions, and a size-appropriate polar body [23].

The assessment of embryo quality is based on its rate of development and morphological features evaluated by light microscopy. The ESHRE Atlas of Human Embryology provides reference images of oocyte and embryo development [25]. Consensus points in the scoring system for blastocysts include the stage of development of the embryo, the grade of the inner cell mass, and the grade of the trophectoderm. A top-quality embryo comprises an expanded blastocyst, with an inner cell mass that is easily discernible, having many cells that are compacted and tightly adhered together, and a good trophectoderm with many cells forming a cohesive epithelium [23].

## 2. Methods

We searched PUBMED for articles published in the English language between September 1991 and June 2022 using the following MeSH search terms: ‘endometriosis’ combined with ‘oocytes’ OR ‘embryonic structures’ OR ‘in vitro fertilization’ OR ‘assisted’ AND ‘fertilization’ with restriction to the human species, no other exclusions were applied. References of the selected articles were screened to identify relevant studies. Data were extracted independently by two investigators (SL and ES).

## 3. Oocyte Quality in Women with Endometriosis

Oocyte quality is reflected by oocyte morphology and the ability of an oocyte to complete maturation and fertilization. In a meta-analysis evaluating the potential effects of oocyte morphological abnormalities on IVF outcomes, fertilization rate was found to be significantly associated with four oocyte morphological abnormalities: first polar body enlargement, perivitelline space enlargement, and the presence of refractive bodies and intracytoplasmic vacuoles [26].

A systematic review of the literature showed that oocytes retrieved from women affected by endometriosis are more likely to fail in vitro maturation (IVM), have altered morphology, and have a lower cytoplasmic mitochondrial content compared to women with other causes of infertility [10]. A study evaluating oocytes that underwent IVM from women with and without endometriosis demonstrated that a significantly lower number of immature oocytes subjected to IVM reached metaphase II (MII) in the endometriosis group compared to controls [27]. IVM oocytes from patients with endometriosis had a significantly higher incidence of cortical granule loss, spindle disruption, and zona pellucida hardening, which could affect oocyte fertilization [27]. Similarly, a retrospective analysis of 1568 mature oocytes in women with endometriosis compared to healthy controls showed a significant increase in morphological abnormalities of the cytoplasm, the zona pellucida and the first polar body [28]).

Studies of IVF outcomes using donated oocytes from healthy women transferred to women with endometriosis have shown that recipients have implantation and pregnancy rates equivalent to patients without endometriosis, supporting the hypothesis that endometriosis is associated with poor oocyte quality [3,5,29]. In women whose donated oocytes came from women with Stage III and IV endometriosis, implantation rates were found to be significantly reduced [29]. In view of the lower implantation and clinical pregnancy rates in women with endometriosis undergoing IVF using their own oocytes [15], these findings support the concept that women with endometriosis have lower oocyte quality and normal endometrial receptivity.

However, a number of publications suggest that there is no reduction in oocyte quality in women with endometriosis. In a large retrospective study of 596 women, the presence of deep pelvic endometriosis lesions or endometriomas did not have an adverse impact on oocyte morphology [30], although women with endometriosis had fewer metaphase II oocytes retrieved, a lower total number of embryos and a number of top-quality embryos and a lower cumulative clinical pregnancy rate, which the authors attributed to lower oocyte yield. A recent prospective cohort study of 503 IVF cycles comparing women with endometriosis to controls with tubal or unexplained infertility reported no difference in the mean fertilization rate, independent of age and r-ASRM classification [31], findings that are consistent with several previous studies [20,32,33,34].

There are several mechanisms postulated for the possible lower oocyte quality in women with endometriosis. The presence of ovarian endometriosis is proposed to induce local pelvic inflammation and higher oxidative stress in the ovarian cortex, resulting in altered follicular and oocyte development and altered oocyte morphology [35,36]. Reactive oxygen species (ROS) have been suggested to reduce oocyte quality by promoting meiotic abnormalities and chromosomal instability in women with endometriosis [37]. Moreover, the follicular milieu in infertile women with endometriosis shows a lower antioxidant capacity and increased oxidative stress on mass spectrometry [38,39]. Iron-mediated oxidative damage to the surrounding follicles has been associated with the presence of ovarian endometriosis, with higher levels of iron in the fluid of follicles developing adjacent to an endometriotic cyst compared to contralateral healthy ovaries [40]. In addition, RNA sequencing of oocytes in women with ovarian endometriosis and healthy oocytes has shown that women with endometriosis have a differential transcriptomic profile associated with lower oocyte quality [41]. Affected pathways included those related to key biological processes such as steroid metabolism, response to oxidative stress, and regulation of cell growth regulation, which could explain the reduction in oocyte quality [41].

The differing parameters used to assess oocyte quality make it challenging to compare findings between studies and may explain the conflicting data in the literature. The majority of studies have been conducted in small samples, and each investigated a limited number of varying morphological abnormalities. Several studies in the literature provide an indirect assessment of the effect of endometriosis on oocyte quality by studying granulosa cells or follicular fluid content of women with endometriosis. Others have used embryo quality as an indicator of oocyte quality, which is influenced by the male partner [42,43].

The development of an updated consensus on oocyte and embryo quality assessment parameters aims to address this for future research studies. Current evidence suggests that the clinical impact of oocyte morphology on reproductive outcomes in women with endometriosis undergoing IVF is limited. While there is evidence to suggest that impaired oocyte morphology in women with endometriosis may have an adverse impact on fertilization rate, most studies have shown that there is no difference in embryo cleavage, implantation, or clinical pregnancy rates following IVF, suggesting that once the oocyte is fertilized, the chance of pregnancy is similar in women with and without endometriosis [20,30,31,32,33,44].

## 4. Embryo Quality in Women with Endometriosis

Several studies have examined the quality of embryos derived from the oocytes of women with endometriosis to determine the impact of endometriosis on embryo quality, with conflicting results. A systematic review and meta-analysis evaluating the impact of endometriosis on embryo quality suggested that endometriosis does not compromise embryo quality from the perspective of morphology, with authors highlighting that there is a need for universal criteria and terminology for grading embryos to reduce the large heterogeneity between studies [45].

A study of human oocytes exposed to endometriosis fluid from women with stage III/IV endometriosis reported a negative effect on the morphology of embryos with excess cellular fragmentation, proposing that increased cell fragmentation is implicated in impaired embryo development by inducing cell death in surrounding blastomeres or altering blastomere division [46]. Pellicer and colleagues found that women with stage III/IV endometriosis had fewer blastomeres per embryo and a higher number of arrested embryos [6]. Women with endometriosis have also been reported to have an increased incidence of aberrant development of embryos, with more prevalent nuclear and cytoplasmic impairment, cytoplasmic fragmentation, and uneven cleavage in the endometriosis group [47].

Conversely, while women with endometriomas undergoing IVF have been reported to have a significantly lower number of oocytes and number of MII oocytes retrieved compared to controls with tubal and male-factor infertility, there is no difference in their total number of embryos, number of top-quality embryos, clinical pregnancy rate, implantation rate or live birth rate [48]. Women operated for moderate/severe endometriosis are reported to have an equivalent number of cleavage embryos and good-quality embryos compared to women without endometriosis [47]. In a study evaluating the impact of endometriosis on embryonic aneuploidy, analysis of 25,000 blastocysts from women with and without endometriosis undergoing IVF with pre-implantation genetic screening (PGS) showed no difference in aneuploidy rates when stratified for age [49]. Table 1 provides a summary of findings from studies examining the impact of endometriosis on oocyte and embryo quality.

## 5. Endometriosis and IVF Outcome

A large meta-analysis in women undergoing assisted conception for tubal-factor infertility reported that women with ASRM stage I/II endometriosis have reduced fertilization and implantation rates compared to women without endometriosis [14]. Other IVF outcomes, including the number of oocytes retrieved and fertilization rate, were affected by the presence of endometriosis at all stages, suggesting that endometriosis affects fertility, oocyte and embryo quality, and endometrial receptivity [14]. This meta-analysis included articles published prior to 2000 when ART success rates were considerably lower than the current levels.

Another meta-analysis found that women with ASRM stage I/II endometriosis compared to women without endometriosis, had lower fertilization rates with no significant reduction in implantation, clinical pregnancy, or live birth rates [15]. Women with stage III/IV endometriosis were found to have reduced implantation and clinical pregnancy rates but no reduction in live birth rates [15].

A subsequent meta-analysis of women with endometriosis undergoing IVF compared to women without endometriosis similarly reported a reduction in the mean number of oocytes retrieved per cycle and clinical pregnancy rate with no difference in live birth rate [34]. Subgroup analysis showed that women with ASRM stage III/IV had lower clinical pregnancy rates and also lower live birth rates [34].

An analysis of 347,185 IVF cycles from the Society for Assisted Reproductive Technologies (SART) database evaluating the impact of endometriosis on reproductive outcomes also showed that women with endometriosis had a significantly lower oocyte yield, lower implantation rate, and lower live birth rate following IVF treatment [51]. Notably, in this study, women with endometriosis had other co-existing infertility diagnoses. In the small number of women with endometriosis alone, the live birth rate was reported to be similar to or slightly higher than other infertility diagnoses.

A large retrospective study using data from the Latin American Data Registry in patients undergoing IVF showed that endometriosis does not affect the outcome of women having IVF [22]. While women with endometriosis had a lower number of oocytes retrieved and a higher cancellation rate, these findings did not translate to reduced pregnancy and live birth rates [22], findings in keeping with several other meta-analyses [20,48,52,53].

Most recently, 13,614 IVF cycles were analyzed using data from the Human Fertilization and Embryology Authority (HFEA) in women with endometriosis undergoing either donor oocyte recipient or autologous IVF cycles [54]. The authors reported no significant difference in the live birth rate between the two groups in both fresh and frozen embryo transfer cycles, suggesting minimal or no effect of oocyte quality on IVF outcomes in women with endometriosis, though there was no information on the extent of endometriosis or the number of oocytes retrieved.

## 6. Management Options

There are limited studies evaluating medical management options to address the impact of endometriosis on oocyte and embryo quality.

In a Cochrane review evaluating the effect of medical ovarian suppression agents versus no treatment, there was no significant difference in clinical pregnancy rate in women with endometriosis [55]. Importantly, several patients included in this review had previously undergone surgical treatment; therefore, recommendations on ovarian suppression and post-surgical ovarian suppression remain unclear [55].

The effect of gonadotrophin-releasing hormone (GnRH) agonists on oocyte and embryo quality in women with endometriosis has been investigated in a small number of studies. In one study, researchers observed a significantly higher number of oocytes, clinical pregnancy rate, and live birth rate in patients with endometriosis treated with the combination of GnRH agonist and letrozole compared to patients treated with GnRH agonist alone [56]. However, this was a small study, and further evaluation is needed. A Cochrane review evaluating the effectiveness of prolonged GnRH analog use in women with endometriosis prior to IVF reported an uncertain effect on clinical pregnancy, live birth rate, number of oocytes retrieved, and number of embryos due to low-quality evidence, highlighting the need for more data [57].

A few studies have investigated the impact of dienogest on oocyte and embryo quality in women with endometriosis, with varying results. One study showed that women receiving dienogest had significantly higher clinical pregnancy (44.7% vs. 16.7%) and live birth rate (36.8% vs. 11.1%) compared to controls [58]. Treatment with dienogest and the GnRH agonist triptorelin in patients after laparoscopic surgery and before ovarian stimulation in IVF cycles resulted in a significantly higher number of oocytes retrieved and good-quality embryos compared to women receiving no treatment [58]. Several research groups have found no significant impact of dienogest on oocyte quality or fertilization rates in women with endometriosis undergoing IVF [59,60]. Barra and colleagues compared IVF outcomes in patients with endometriosis with and without pre-treatment with dienogest for three months [60]. They found no differences in the number of MII oocytes or embryo quality between groups, although, in women with large endometriomas, dienogest treatment significantly increased the number of oocytes retrieved, two-pronuclear embryos, and blastocysts compared to controls [60]. More recently, a negative effect of dienogest on reproductive outcomes has been suggested. Women with endometriosis taking dienogest prior to IVF were found to have a lower number of mature oocytes retrieved, lower fertilization rates, and lower live birth rates [61,62]. Overall, the evidence on the impact of dienogest on oocyte quality in women with endometriosis is mixed and further research is needed.

Antioxidants have been proposed as a therapeutic approach to improve oocyte quality in patients with endometriosis due to the association between increased oxidative stress and poor oocyte quality. One study found that the administration of vitamins C and E to patients for six months significantly decreased plasma oxidative stress markers, with a trend towards increased pregnancy rates when compared to women receiving a placebo [63]. Porpora et al. reported that N-acetyl cysteine (NAC) supplementation decreases endometrioma size compared to no treatment, with no difference in pregnancy rates [64]. These preliminary findings with antioxidant supplementation need further evaluation.

Pentoxifylline has anti-angiogenic effects and has been postulated to lead to endometriosis lesion regression [65]. Some studies have suggested that pentoxifylline may improve oocyte quality, fertilization rates, and embryo development in women with endometriosis [66,67], whereas others have found no beneficial effect on oocyte and embryo quality [68,69]. The effects of pentoxifylline have been evaluated in a recent Cochrane systematic review based on three randomized controlled trials in 285 patients, which showed that overall the evidence for the effect of pentoxifylline versus placebo on clinical pregnancy rates is uncertain, and there are no trials that have reported on the effects of pentoxifylline on live birth rate [70].

There is limited research on the impact of metformin on oocyte quality in women with endometriosis. Metformin is an anti-inflammatory agent with a modulatory effect on sex-steroid production. Foda and Aal reported that pregnancy rates were significantly higher in patients with Stages I–II endometriosis treated with metformin compared to no treatment (25.71% vs. 11.76%) [71]. Some studies have suggested a positive impact of metformin on oocyte quality, fertilization rate, clinical pregnancy, and live birth rate [72,73], in contrast to others show no difference [74]. Overall, the evidence on the impact of metformin on oocyte quality in women with endometriosis is mixed and further studies are needed to fully evaluate its effect.

A few studies have investigated the role of surgical treatment in women with endometriosis undergoing IVF, though there are no randomized controlled trials. A systematic review and meta-analysis comparing reproductive outcomes in patients who underwent surgery for deep infiltrative endometriosis (DIE) before IVF versus patients who underwent IVF without previous surgery for DIE showed that there might be a beneficial role for surgery prior to IVF treatment, though there is some concern over the accuracy of their analysis and the significant loss to follow up [75]. One research group has compared surgical treatment for minimal and mild endometriosis with diagnostic laparoscopy alone in 661 women prior to undergoing IVF treatment [19]. Notably, the duration of infertility and the time interval between surgery and IVF treatment were substantially different between the groups, which may in itself entirely account for the minor difference in live birth rates (27.7% vs. 20.6%) [19]. Nevertheless, women with ASRM stage I and II endometriosis who underwent surgical removal of all visible endometriosis lesions had significantly improved implantation rates, pregnancy rates, and live birth rates and were noted to have a shorter time to pregnancy and higher cumulative pregnancy rate [19]. Several studies have evaluated the impact of surgery for endometriomas, including a systematic review and meta-analysis by Alborzi and colleagues, reporting no significant difference in pregnancy rate when compared to other treatments such as surgery combined with IVF, IVF alone or aspiration with or without sclerotherapy followed by IVF [76,77]. Table 2 provides a summary of findings from studies evaluating the management options in women with endometriosis and their impact on oocyte and embryo quality.

## 7. Conclusions and Future Research

Endometriosis is a heterogeneous disease associated with infertility. The presence of endometriosis may reduce oocyte quality and embryo quality. However, this does not appear to translate to a clear clinical impact on IVF outcomes. Results from meta-analyses addressing IVF outcomes in affected women suggest that the reduction in the number of mature oocytes retrieved is associated with the presence of endometriosis, while a reduction in fertilization rates is more likely to be associated with minimal/mild rather than with moderate/severe disease [14,15,20,34,52,53]. Overall, there does not appear to be a negative impact of endometriosis on the live birth rate following IVF, despite lower oocyte yield and altered oocyte and embryo quality.

However, further research is needed, particularly regarding the effect of different stages of the disease, its location, and the impact of previous medical and surgical treatment. Surgery in women with endometriosis has not been shown to improve reproductive outcomes following IVF and may further reduce ovarian reserve. Data on the impact of medical treatments for endometriosis on IVF remains limited, and there is a need for well-designed randomized controlled trials to evaluate further the role of surgical and medical treatment options in women with endometriosis undergoing IVF and the impact of the severity of endometriosis and its location.

The development of an updated consensus on oocyte and embryo quality assessment parameters for use in future research studies is fundamental. A greater understanding of the effect of endometriosis on ovarian function will allow better timing and tailoring of medical and surgical treatments. As the field of fertility preservation grows, a greater understanding of the impact of the disease on oocyte quality is fundamental to appropriately counsel suitable patients affected by endometriosis for fertility preservation.

## Figures and Tables

**Table 1 jcm-12-04186-t001:** Summary of findings from studies examining the impact of endometriosis on oocyte and embryo quality.

Study Authors	Study Design	Impact of Endometriosis on Oocyte/Embryo Quality	Study Findings
Goud et al., 2014 [27]	Prospective cohort study, *n* = 28 women	Oocyte quality reduced ↓	- Increased likelihood of oocyte to fail in vitro maturation IVM; - Altered oocyte morphology (cortical granule loss, spindle disruption, zona pellucida hardening).
Kasopoglu et al., 2017 [28]	Retrospective cohort study, *n* = 72 women	Oocyte quality reduced ↓	- Altered oocyte morphology (morphological abnormalities of the cytoplasm, zona pellucida, and first polar body).
Simon et al., 1994; Sung et al., 1997; Diaz et al., 2000 [3,5,29]	Restrospective cohort study, *n* = 137 women; retrospective cohort study, *n* = 239 women; matched case-control study, *n* = 58 women	Oocyte quality reduced ↓	- Lower implantation rates in donor oocytes from women with endometriosis; - Equivalent implantation rates and pregnancy rates when women with endometriosis using donor oocytes from healthy women.
Ferrero et al., 2019 [41]	Prospective cohort, *n* = 12 women	Oocyte quality reduced ↓	- Differential transcriptomic profile associated with lower oocyte quality.
Sanchez et al., 2017 [10]	Review article	Oocyte quality reduced ↓	- Altered oocyte morphology; - Increased likelihood of oocyte to fail IVM; - Lower cytoplasmic mitochondrial content.
Robin et al., 2021 [30]	Retrospective cohort study, *n* = 596 women	Oocyte quality unaffected ↔	- Normal oocyte morphology; - Lower number of top-quality embryos and lower cumulative clinical pregnancy rate are both attributed to lower oocyte yield.
Metzemaeker et al., 2020, Filippi et al., 2014, Yang et al., 2015, Hamdan et al., 2015 [20,31,32,34]	Population-based cohort study, *n* = 503 IVF cycles [31]; prospective cohort study, *n* = 29 women [32], systematic review and meta-analysis, *n* = 1039 women [33], systematic review and meta-analysis, *n* = 928 women [35]	Oocyte quality unaffected ↔	- Equivalent fertilisation rates.
Brizek et al., 1995 [50]	Retrospective cohort study, *n* = 235 embryos	Embryo quality reduced ↓	- Increased incidence of aberrant development of embryos (more prevalent nuclear and cytoplasmic impairment, cytoplasmic fragmentation, uneven cleavage).
Pellicer et al., 2001 [6]	Retrospective cohort study, *n* = 70 women	Embryo quality reduced ↓	- Altered embryo morphology (fewer blastomeres per embryo, a higher number of arrested embryos.
Paffoni et al., 2019 [46]	Randomized-controlled in vitro study, *n* = 147 oocytes	Embryo quality reduced ↓	- Altered embryo morphology (excess cellular fragmentation, cell death in blastomeres, and altered blastomere division.
Alshehre, et al., 2020 [48]	Systematic review and meta-analysis, *n* = 8 studies	Embryo quality unaffected ↔	- No difference in total number of embryos; - No difference in number of top-quality embryos; - No difference in clinical pregnancy rate, implantation rate, or live birth rate.
Sanchez et al., 2020 [47]	Retrospective matched cohort study, *n* = 3818 embryos	Embryo quality unaffected ↔	- Equivalent number of cleavage embryos; - Equivalent number of good-quality embryos.
Dongye et al., 2021 [45]	Systematic review and meta-analysis, *n* = 22 studies	Embryo quality unaffected ↔	- Normal embryo morphology.
Juneau et al., 2017 [49]	Retrospective cohort study, *n* = 305	Embryo quality unaffected ↔	- No difference in aneuploidy rates.

**Table 2 jcm-12-04186-t002:** Summary of findings from studies evaluating management options in women with endometriosis and their impact on oocyte and embryo quality.

Study Author(s)	Type of Study	Management Option for Endometriosis	Study Findings
Hughes et al., 2007 [55]	Cochrane review, *n* = 24 studies	Medical ovarian suppression agents (including danazol, progestins, and oral contraceptives)	- No impact on pregnancy rates.
Georgiou et al., 2019 [57]	Cochrane review, *n* = 640 women	Prolonged GnRHa use prior to IVF	- Uncertain effect on clinical pregnancy, live birth rate, number of oocytes retrieved, and number of embryos.
Kim et al., 2017 [59]	Retrospective cohort study, *n* = 64 women	Letrozole	- No impact on number of oocytes, number of embryos, fertilization rates.
Cantor et al., 2019 [56]	Retrospective cohort study, *n* = 126 women	GnRH agonist alone vs GnRHa + Letrozole prior to IVF	GnRHa + Letrozole - Higher number of oocytes retrieved; - Higher clinical pregnancy rate; - Higher live birth rate.
Muller et al., 2017 [58]	Prospective cohort study, *n* = 144 women	Dienogest vs. GnRHa vs. control following laparoscopic surgery prior to IVF	Dienogest, GnRHa - Higher number of oocytes retrieved; - Higher number of good-quality embryos. Dienogest - Higher clinical pregnancy rates; - Higher live birth rates. GnRHa - No difference in clinical pregnancy rates and live birth rates.
Barra et al., 2020 [60]	Retrospective analysis of prospectively collected data, *n* = 151 women	Pre-treatment with Dienogest for three months	- No difference in number of MII oocytes or embryo quality; - Increased number of oocytes retrieved and number of embryos in women with large endometriomas.
Mier-Cabrera et al., 2008 [63]	Observational cohort study, *n* = 34	Vitamin C and E for six months	- Decrease in plasma oxidative stress markers; - Trend towards increased pregnancy rates.
Porpora et al., 2013 [64]	Observational cohort study, *n* = 92	N-acetyl cysteine (NAC) supplementation	- Decrease in endometrioma size; - No difference in pregnancy rates.
Grammatis, et al., 2021 [70]	Cochrane review, *n* = 285 women	Pentoxifylline	- Uncertain impact on pregnancy rates.
Foda and Aal, 2012 [71]	Observational cohort study, *n* = 69 women	Metformin	- Higher pregnancy rates.
Casals et al., 2021 [75]	Systematic review and meta-analysis, *n* = 5 studies	Surgery for DIE prior to IVF vs. no surgery	- Higher pregnancy rates.
Opoien et al., 2011 [19]	Retrospective cohort study, *n* = 661 women	Surgical removal of all visible endometriosis lesions in stage I-II endometriosis vs. diagnostic laparoscopy	- Higher implantation rate, pregnancy rate and live birth rate (27.7% vs. 20.6%) - Shorter time to pregnancy.
Alborzi et al., 2019 [76]	Systematic review and meta-analysis, *n* = 553 women	Comparison of four different treatment options for endometriomas: Surgery for endometrioma + IVF vs. surgery for endometrioma + spontaneous pregnancy vs. aspiration ± sclerotherapy + IVF vs. IVF alone	- No difference in pregnancy rates.
Demirdag et al., 2021 [77]	Retrospective cohort study, *n* = 1936 women	Surgery for endometrioma vs. no surgery for endometrioma vs. control	- No difference in pregnancy rates.

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
