# Peer review of "Endometriosis, Oocyte, and Embryo Quality"

_jcm, 2023, doi:10.3390/jcm12134186_

Round 1

Reviewer 1 Report (Previous Reviewer 4)

The authors have adequately addressed the comments and have improved the overall quality of the manuscript. 

Author Response

Reviewer's comment: The authors have adequately addressed the comments and have improved the overall quality of the manuscript. 

Author's response: Thank you for this feedback.

Reviewer 2 Report (Previous Reviewer 3)

most revision are adequate and appropriated but the table is not informative. i would suggest to prepare a better one in more details. in addition another table to summarise the available treatment options may be necessary too.

Author Response

Reviewer's comment: most revision are adequate and appropriate but the table is not informative. i would suggest to prepare a better one in more details. in addition another table to summarise the available treatment options may be necessary too.

Author's response: thank you for this feedback. We have now updated table 1 in the manuscript to make it more detailed and informative. We have also included an additional table in the manuscript (table 2) summarising the evidence for the impact of different management options on oocyte and embryo quality.

Reviewer 3 Report (Previous Reviewer 2)

Congratulations to the authors for the improvement in the manuscript.

Table1 is a great contribution to the paper, but it could be improved. Title should be in the top. The structure could be improved, for example with a column to the type of study, it could be divided to oocytes and embryos, it could be more clear if it is reported an impact or not in oocyte or embryo quality, etc.

I suggest a table also for "Endometriosis and IVF outcome"

Globally it is lacking a summary of the evidence found by the authors in each subchapter and information regarding the type of endometriosis analyzed in each study. Is it different in superficial, deep or ovarian endometriosis?

Finally, in "management options", the description of the surgical option is very succinct, namely the  impact on ovarian reserve, oocyte quality, embryo quality or IVF.

Author Response

Reviewer's comment 1: Congratulations to the authors for the improvement in the manuscript.

Author's response to reviewer's comment 1: Thank you for this feedback

Reviewer's comment 2: Table1 is a great contribution to the paper, but it could be improved. Title should be in the top. The structure could be improved, for example with a column to the type of study, it could be divided to oocytes and embryos, it could be more clear if it is reported an impact or not in oocyte or embryo quality, etc.

Author's response to reviewer's comment 2: As advised, we have now updated table 1 in the manuscript to include a column for study design, divided into oocytes/embryos and made it more clear as to whether there is an impact on oocyte/embryo quality.

Reviewers comment 3: I suggest a table also for "Endometriosis and IVF outcome"

Author's response to comment 3: Thank you for this suggestion. Based on overall reviewer's responses we have now included a second table summarising the evidence for the impact of different management options on oocyte and embryo quality. We considered incuding a further table summarising the impact of endometriosis IVF outcome, however we found this to be outside the scope of this particular review. 

Reviewer's comment 4: Globally it is lacking a summary of the evidence found by the authors in each subchapter and information regarding the type of endometriosis analyzed in each study. Is it different in superficial, deep or ovarian endometriosis?

Author's response to reviewer's comment 4: Thank you for this question. Most available studies on oocyte and embryo quality target endometriosis as a single distinct disorder with no differentiation between ovarian endometriosis and other forms of the disease so we are unable to draw form conclusions based on the type of endometriosis.

Reviewer's comment 5: Finally, in "management options", the description of the surgical option is very succinct, namely the  impact on ovarian reserve, oocyte quality, embryo quality or IVF

Authors response to reviewer's comment 5: thank you or this suggestion. We have now included a further table summarising the evidence for the impact of different management options on oocyte and embryo quality including surgery and hope this will suffice. An exhaustive review of the impact of surgery for endometriosis on reproductive outcomes is outside the scope of this particular review. 

This manuscript is a resubmission of an earlier submission. The following is a list of the peer review reports and author responses from that submission.

Round 1

Reviewer 1 Report

I have read the manuscript with interest. The paper reviews the current literature on the impact of endometriosis on oocyte-embryo quality and IVF outcome as well as the medical and surgical management strategies. The topic is interesting and has been a long-standing matter of debate. Authors make a brave attempt to unravel some of the controversial issues. However, they are constrained by the quality of published data.

 The manuscript is well written and clear. Authors cover and interpret the available data and honestly assess their strengths and limitations.

 I have just a single comment/suggestion. Authors might consider changing the order of topics to be discussed, start with the impact on oocyte and embryo quality and then move to endometriosis and IVF outcome.

Reviewer 2 Report

OVERALL ANALYSIS

Thanks for the opportunity to review this manuscript. First, I would like to congratulate the authors for choosing the topic, which is extremely interesting from a clinical and scientific point of view.

The manuscript addresses an enigmatic and complex disease - endometriosis, that is frequently associated with infertility. Thus, reviewing the impact of the disease in oocyte and embryo quality is extremely relevant.

However, there are some issues that should be reviewed.

ESPECIFIC ANALYSIS

Keywords - Choose MESH terms. For example, change “assisted conception” to “Assisted Reproductive Technics”

Introduction - In the end of the introduction the authors define the aim of the study, and it seems that they will just focus ovarian endometriosis.

Methods

-        It is not clear how the authors select the manuscripts. Where they excluded studies including patients with previous surgery?

-        Also, ‘embryo quality’, ‘oocyte quality’, ‘IVF’, ‘assisted conception’ and ‘ART’ are not MESH terms.

Endometriosis and IVF outcome

-        This chapter has small number of references for what is found in literature. Also the authors do not summarize the existent evidence, only cite the results of each study individually.

-        For example, the authors cited 6 papers, being 3 metanalysis and 3 clinical studies. However, in a rapid search in Pubmed, there are some systematic reviews and meta-analysis that were not included, as:

-          Yang C, et al. Impact of ovarian endometrioma on ovarian responsiveness and IVF: a systematic review and meta-analysis. Reprod Biomed Online. 2015

-          Barbosa MA, et al. Impact of endometriosis and its staging on assisted reproduction outcome: systematic review and meta-analysis. Ultrasound Obstet Gynecol. 2014

-          Qu H, et al. The effect of endometriosis on IVF/ICSI and perinatal outcome: A systematic review and meta-analysis. J Gynecol Obstet Hum Reprod. 2022

-          Dongye H, et al. The Impact of Endometriosis on Embryo Quality in in-vitro Fertilization/Intracytoplasmic Sperm Injection: A Systematic Review and Meta-Analysis. Front Med (Lausanne). 2021

Assessment of oocyte and embryo quality - This chapter is not within the scope defined by the authors or even related to the type of research referred to in the methods. Therefore, it may eventually be equated in the introduction, in the context of the study theme.

Oocyte quality in women with endometriosis

-        This chapter is well prepared, as at the end it explains the mechanisms, the limitations and proposes an explanation for this issue.

-        However, they seem too low the number of reported studies and it would be interesting that the authors explain better each study and also show if there is some difference regarding the endometriosis stage and if it is ovarian endometriosis or not.

Embryo quality in women with endometriosis

-        This chapter could also be better designed, with at least a summary of the existing evidence.

-        Alshehre, et al., 2020 is a study that also reports data on oocytes not mentioned in the previous chapter, in addition, they specifically studied in patients with ovarian endometriosis that it would be interesting to discuss in the chapter “Oocyte quality in women with endometriosis”.

Management options

-        This is a very interesting chapter from a clinical point of view. However, and especially for the surgical option, the references cited by the author are very scarce. For example, for endometrioma surgery, they only have one article and there are many studies in this area (for example, DOI: 10.3390/biomedicines11030844, 10.23736/S2724-606X.22.05188-0, 10.1016/j.ejogrb.2021.06.034).

-        Additionally, the impact on ovarian reserve, oocyte quality, embryo quality or IVF could be also analysed regarding the type of surgery and that is not mentioned.

Conclusion - References are missing in the sentence of lines 250-253.

Reviewer 3 Report

The authors submit a manuscript named “Endometriosis, oocyte and embryo quality”. This review is generally interesting. By evaluating the current literature, the authors proposed the effect of endometriosis on fertility, the potential mechanisms of the adverse effect, and relevant management options. It indicates that the presence of endometriosis evidently reduces ovarian reserve, oocyte quality and embryo quality, while more well-designed clinical trials are needed for further evaluation.

There has been a great load of work here. However, there are several comments which authors should consider addressing:

1.      Introduction:

P1L22: Since there are different subtypes of endometriosis, the underlying mechanisms of infertility induced by each subtype of endometriosis should be distinct. More elaborations are needed here.

P1L23 ‘Proposed theories include mechanical distortion of pelvic anatomy, impairment of gamete and embryo 23 transport, reduced endometrial receptivity and reduced oocyte and embryo quality’: There are several proposed theories of endometriosis-related infertility, why only focused on impaired oocyte quality, embryo quality, and endometrium receptivity but no description of other mechanisms? If this manuscript intends to reveal the correlation between endometriosis and oocyte quality, embryo quality, and endometrium receptivity during ART application, then more elaborations are needed, and the title should be amended accordingly.

P1L29 ‘Several studies describe a negative effect of endometriosis on the reproductive outcomes of women undergoing IVF in contrast to others which show no difference in outcomes’: What are the ‘others’? Could you list some example diseases which show no effect on the reproductive outcomes of women undergoing IVF?

P2L32 ‘The reasons for these observed differences and whether ovarian endometriosis affects oocyte quality, embryo quality and endometrial receptivity have not yet been clearly determined’: The previous description is about endometriosis in general, while it points out the subtype of ovarian endometriosis suddenly here, therefore either add the illustration of other subtypes or replace the sentence with the reason for endometriosis-related infertility generally.

2.      Method:

P2L39: For the searching terms, why only include ‘endometriosis’ and ‘endometrioma’ while no other subtypes i.e. ‘peritoneal endometriosis’ and ‘deep infiltrating endometriosis’. If you want to focus on endometrioma, more elaborations are needed in introduction.

3.      Results:

P9L177 Management options:

It would be clearer if you can make a table to compare the pros and cons of different management options.

Before introducing the therapeutic options that are still under investigation, it’s better to mention the most common and conventional management options for endometriosis-related infertility in clinic first.

4.      Reference:

The references are a bit out of date, esp. for those have been published more than 10 years ago. More updated literatures are needed.

Reviewer 4 Report

The review is well-written throughout and includes a very elaborate compilation of information on impact of endometriosis on oocyte and embryo quality. However, I have few concerns to be addressed for better understanding and flow. 

1. Authors are suggested to reframe or write the first paragraph in the introduction part since sentences from abstract has been repeated.

2. The aim of the review seems very superficial , it needs further elaboration and emphasis. 

3. Expansion for acronyms could be provided at the first place (eg: ESHRE and ASRM etc)

4. The authors are suggested to move the "Oocyte quality assessment" in the methods section since it disrupts the flow. 

5. In line 103, page no 5 , authors have used recent for a publication of 2017. kindly remove that.  

6. Many recent studies evaluating the association of endometriosis and IVF outcomes have been missed out. The authors need to include more recent studies to the manuscript.   

7. The conclusion does not do the justice of conveying the key points from the review. The authors have repeated some of the finding of previous studies in the conclusion. 

8. The results from different study could be presented as schematic diagram or tables for easy understanding of the audience.